# Recent Developments of Electrochemical and Optical Biosensors for Antibody Detection

**DOI:** 10.3390/ijms21010134

**Published:** 2019-12-24

**Authors:** Wei Xu, Daniel Wang, Derek Li, Chung Chiun Liu

**Affiliations:** 1Department of Biomedical Engineering, Case Western Reserve University, Cleveland, OH 44106, USA; wxx104@case.edu; 2Department of Chemical and Biomolecular Engineering, Case Western Reserve University, Cleveland, OH 44106, USA; dnw17@case.edu; 3Solon High School, Solon, OH 44139, USA; dxl670@case.edu

**Keywords:** antibody detection, electrochemical detection, optical detection, biosensors

## Abstract

Detection of biomarkers has raised much interest recently due to the need for disease diagnosis and personalized medicine in future point-of-care systems. Among various biomarkers, antibodies are an important type of detection target due to their potential for indicating disease progression stage and the efficiency of therapeutic antibody drug treatment. In this review, electrochemical and optical detection of antibodies are discussed. Specifically, creating a non-label and reagent-free sensing platform and construction of an anti-fouling electrochemical surface for electrochemical detection are suggested. For optical transduction, a rapid and programmable platform for antibody detection using a DNA-based beacon is suggested as well as the use of bioluminescence resonance energy transfer (BRET) switch for low cost antibody detection. These sensing strategies have demonstrated their potential for resolving current challenges in antibody detection such as high selectivity, low operation cost, simple detection procedures, rapid detection, and low-fouling detection. This review provides a general update for recent developments in antibody detection strategies and potential solutions for future clinical point-of-care systems.

## 1. Introduction

Rapid, accurate, simple, and cost-effective disease diagnostic methods have progressed significantly, including personalized medicine in the medical industry [1]. The growing understanding of disease pathological pathways provides different biomolecules as the indicators for various diseases. These biomolecules are generally defined as the biomarkers of the disease and quantitative information from such biomarkers assists in assessing the stage of the disease and providing information to design effective drug treatments, leading to the development of personalized medicine [2,3]. Various biomarkers, including proteins, small molecules, and nucleic acids, are efficacious indicators for cancers, neuro-degenerative disorders, and genetic disorders [4,5,6,7,8]. 

Among the biomarkers of interest, antibodies represent an important type of detection marker for several reasons. First, for many human diseases, such as oncological, inflammatory, neurological and psychiatric disorders, the human immune system defends against external antigens through the production of the corresponding antibodies. Therefore, quantification of the antibody is a reliable indicator for disease progression [9]. Second, as therapeutic antibodies are established as a successful class of drugs to specifically target cancers and inflammatory diseases, quantification of the antibodies is critical for clinically tracking the toxicity of the drug treatment and assessing the efficiency of the drug delivery [10]. Third, antibodies are abundant in human blood; consequently, they provide a feasible and reliable opportunity for biomarker detection, owing to the simplicity of blood processing [11]. Moreover, the challenge for antibody detection is actually diminished by the evolutionary process. The production of antibody is the result of prior expression of the corresponding antigen by the disease-related genome. Therefore, nature provides a high-affinity antibody binding agent, the antigen, which is used as the recognition element for the antibody detection, predigesting the efforts for specific design of the recognition elements for biosensing systems. Currently, the conventional methods for detection of such biomarkers, such as ELISA and Western Blot, require exhaustive processing steps and multiple reagents, which are expensive, difficult to operate, and time-consuming [12]. Thus, these methods are not ideal for the robust demands of a rapid, simple, cost-effective point-of-care system. In order to overcome this concern, a substantial number of advanced detection methods of antibodies have recently been developed, especially methods based on optical and electrochemical systems [13,14,15,16]. Optical systems measure an optical signal through fluorescent, colorimetric, or luminescent methods, enabling rapid, sensitive, and direct detection of antibodies. Electrochemical systems are also ideal for the development of point-of-care devices due to their simplicity, portability, rapidness, cost-effectiveness and sensitivity with the capability of applying various transduction methods [14]. Detection of target biomarkers directly using human sera samples can be achieved using electrochemical detection method [17,18].

However, technical challenges remain for both the electrochemical and the optical biosensing systems in antibody detection. For example, the matrix effect induced by the non-specific components in complex biological fluids causes nonspecific adsorption leading to false-positive results. The sensitivity and the detection limit of antibodies are also critical for an early disease diagnosis, due to the relatively low abundance of antibodies in human fluids at the early stage of the disease. Furthermore, in order to improve the detection efficiency, the antibody detection process needs to be rapid. For a globally applicable point-of-care system, the detection strategy should also be portable and cost-effective. In this review, we specifically select the few recent advances of antibody detection strategies that can be representative, which resolve the obstacles described above, delivering potential clinically applicable platforms. Detailed descriptions of detection strategies and future perspectives for the ultimate point-of-care systems are discussed. This review aims to provide the biosensing community a general update on trends in antibody detection strategies and the potential general methods for clinical applications. 

## 2. Electrochemical Detection

Electrochemical-based biosensors quantify the antibody concentration by measuring the current or the impedance through various electrochemical analytical methods. Electrochemical biosensors provide simple, cost-effective, and accurate measurements [14]. However, challenges remain in directly employing electrochemical detection strategies for point-of-care systems. First, selectivity of the detection methods, which can differentiate different antibodies, is crucial for the reliability of the detection results. This challenge can be addressed by using high-affinity antibody recognition elements, such as epitope or antigen. Second, high sensitivity is necessary to provide a reliable and generalized detection platform. With the development of simple fabrication methods of highly sensitive materials [19,20,21,22,23], the sensing material can provide a precise response to minute changes in electrical conductivity. Moreover, the simplicity of the detection strategy is critical for its potential applications. The sensing strategy can be simplified by creating a novel sensing platform that is non-label, and reagent-free [24,25,26]. Reducing fouling events is also a key challenge for antibody detection since fouling can lead to false-positive detection signals [24,27]. Therefore, construction of an anti-fouling electrochemical surface is important to enhance the detection accuracy. 

### 2.1. Selective Detection Strategy

A key challenge in the development of antibody detection platforms for point-of-care use is selectivity. Highly specific, target responsive, biomolecular scaffolds demonstrate an excellent selectivity for antibody detection. An antibody-activated electrochemical “switch” was recently demonstrated with a high detection specificity (Figure 1a) [28]. Bound by thiol group to a gold working electrode, the platform’s recognition element consisted of a single-stranded DNA probe containing two target recognition elements (antigens, epitopes, or haptens). The transduction mechanism was based on a proximity change between the sensing element and a methylene blue tag. When target antibodies were not present, the DNA probe adopted a stem-loop conformation and brought the redox label close to the gold electrode, enabling a rapid electron transfer rate and generating a significant Faradaic current. In the presence of target antibodies, the DNA probe unfolded, creating a longer distance between the tag and the electrode, resulting in a low electron transfer rate and a low Faradaic current. This platform for antibody detection was insensitive to nonspecific adsorption and enhanced selectivity because the recognition element was associated with a large-scale conformational change. The change was activated by a high binding affinity that was produced by specific binding of a target antibody. This strategy was applied for the detection of anti-DNP, digoxin antibody, and anti-HIV antibody AF5, and utilized different recognition elements such as small-molecule haptens and polypeptide epitopes, achieving a nanomolar level of detection within 10 min in whole blood.

Similarly, Wei et al. also developed a strategy to quantify antibody in serum using a DNA molecular switch (Figure 1b) [29]. The recognition element was the “capture” DNA complex with two digoxin targets on either end of a single strand DNA sequence. The electrochemical signal of ferrocene based on its position to the electrode served as the transduction element. The sensing element was a thiol-modified hairpin DNA structure labeled with ferrocene bound to an electrode. Incubation of the capture DNA allowed the six cytosine bases to interact with the complementary sequence located in the hairpin DNA structure, forming the “triplex-stem” DNA structure. The formation of the triplex-stem DNA separated the ferrocene on the hairpin DNA structure away from the electrode, reducing the electrochemical signal. Digoxin antibody bound to the digoxin in the capture DNA sequence released the capture DNA, leaving only the hairpin DNA on the electrode and returning the ferrocene closer to the electrode. This method demonstrated a detection range of 1.0–500 pg/mL and a lower detection limit of 0.4 pg/mL (5.8 fM). Dection of digoxin antibodies in blood serum was achived with 91.6% to 105% recovery rate. Selectivity of the sensor was also demonstrated with less than 15% recovery effiency of antibodies other than target digoxin.

A carbon electrode modified with carboxyl nanotubes was used to detect antibodies of Zika virus (ZIKV), Dengue virus (DENV), Circumsporozoite protein (CSP) 210 and CSP 240. Cabral-Miranda et al. developed a sensor to distinguish between ZIKV antibodies and DENV antibodies in blood and saliva after the recent outbreak of ZIKV (Figure 2a) [30], and the cross-reactivity between the antibodies. The recognition elements were the envelope protein domain III (EDIII) and non-structural protein 1 (NS1) proteins from residues 299–407a of ZIKV. Electrochemical impedance spectroscopy (EIS) was the transduction mechanism measuring the resistance within the biosensor through a K_3_Fe(CN)_6_ and K_4_Fe(CN)_6_ redox solution. A three-electrode system of carbon electrodes with silver connections served as the sensing element of the biosensor. The proteins were immobilized to an activated surface of the carbon working electrode. In pure sera, this sensor detected antibodies at a lower limit of 53 fg/mL and 17 fg/mL for the EDIII and NS1 based sensors, respectively. The sensors were able to selectively recognize antibodies against the two proteins in both human sera and saliva. When testing for cross-reactivity, the ZIKV antibodies did recognize the biosensor, but the DENV antibodies did not. 

Cardoso et al. also investigated an immobilization method for the biosensor in the detection of the CSP antibodies during infection in malaria diagnosis (Figure 2b) [31]. The recognition element was a peptide with antigen regions CSP 210–247, which was immobilized onto the carbon nanotube (CNT) surface after modifications to activate and aminate the carboxylic surface. The addition of the antigen onto the electrode and CNT increased the charge-transfer resistance of the overall system. The electrochemical responses were measured through EIS. A carboxylated carbon nanotube monolayer was used to coat the carbon electrode surface of the biosensor, increasing the electrical properties of the layer. Both CSP 210 and CSP 247 antibodies were tested individually and also in combination in mouse sera, showing selective detection of Anti-CSP 210–247. The detection limit of this biosensor was in the range of 50 fg/mL. This strategy indicated a potential early detection method for diseases by creating a single sensor covering a group of antibodies of the same or different disease.

These strategies demonstrate the selective detection of selected antibodies. The conformational changes in DNA create more selective sites enhancing the detection. Furthermore, the use of DNA bypasses the need to use additional labeling and is highly associated to the target in contaminant-rich specimens such as serum. Finally, distinct protein sequences create the ability to detect individual antibodies between similar families of viruses. The applications targeted here, flaviviruses and malaria are a small sample, but indicate the value that such a selective detection process can provide as point-of-care systems for tracking a disease. 

### 2.2. Simplified Detection Strategy

A point-of-care system is necessary in resource-limited areas. Thus, simplicity for both the biosensor fabrication and the detection procedures is critical [32]. It is critical to create a reagent-free electrochemical platform which is easy to operate. A novel sensing platform based on the antibody-catalyzed water oxidation pathway (ACWOP) sensing platform was created as a simplified detection method (Figure 3a) [33]. Because single antibodies naturally catalyzed the production of hydrogen peroxide (H_2_O_2_) in the presence of water and a singlet oxygen (O_2_*), this strategy enables antibody quantification through the detection of H_2_O_2_. The recognition element consisted of haptens bound to poly(oligoethylene glycolmethacrylate) (POEGMA) polymer brushes. The transduction mechanism was based on the redox response of resorufin (7-hydroxy-3H-phenoxazin-3-one). The sensing element was a glassy carbon working electrode in a three-electrode system. The platform used DNP haptens to detect anti-DNP, and was capable of detecting H_2_O_2_ at concentration as low as 0.33 nM. Additional tests indicated that the device could detect less than 3 pg antibodies in a 10 μL sample (2 pM). This strategy allowed for the direct detection of antibodies, via H_2_O_2_, regardless of the specificity of the antibody, eliminating the need for specially prepared secondary reagents, and reducing the complexity of the preparation process. 

A redox probe is commonly used to enable changes on the biosensor electrode surfaces for quantification of antibodies with electrochemical analytical techniques [34,35,36,37]. Thus, quantum dots (QDs) were utilized for the detection of antibodies (Figure 3b) [38]. For this design, the recognition element was enzyme-based and this platform was modeled with tissue transglutaminase (tTG) for the detection of anti-tTG IgG antibodies. The transduction element consisted of an acid attack on the quantum dots. This platform was modeled with dSe/ZnS QDs and hydrochloric acid for the acid attack. Electrochemical detection of Cd^2+^ released from the QDs was directly correlated to the antibody concentration. This procedure was performed on a sensor surface formed by eight three-electrode electrochemical cells with carbon-based working and counter electrodes. Normally, the bioassay and the subsequent acid attack were performed outside of the detection platform. However, the acid attack and the detection were performed directly on the screen-printed electrodes for this platform. This strategy uses QDs as a simple approach for the detection of antibodies.

Many current biosensor platforms rely on externally labeled biomolecules in order to process a transducing signal [39,40,41,42,43]. A non-label platform was developed to simplify the detection process for pathogens like hantaviruses (Figure 3c) [44]. The recognition element was a target specific antigen—HNp. The sensor surface was prepared with functionalized self-assembled monolayers (SAM) on a standard three-electrode system with a gold working electrode. Afterwards, hantavirus Araucaria nucleoprotein (HNp) was immobilized to the sensor through an EDC/NHS reaction for specific detection of anti-hantavirus antibodies, forming a Au-MPA-HNp complex. This model was a simple detection platform created for anti-hantavirus detection and was able detect target antibodies at nanomolar levels within 3 hr. With a rapid and simplified detection system, this platform offers potential to qualify hantavirus detection for point-of-care use in the future. In Figure 1, O_2_* is a singlet oxygen.

These strategies emphasize the simplicity of detection. Current standards are complex and require not only more time, but also more resources. The antibody-catalyzed water oxidation pathway-based platform demonstrates a simplified sensing approach. The QD platform demonstrates an improvement on similar sensing platforms that were previously developed. The hantavirus platform demonstrates the appeal of a non-label method for the transduction mechanism. 

**Figure 3 ijms-21-00134-f003:**
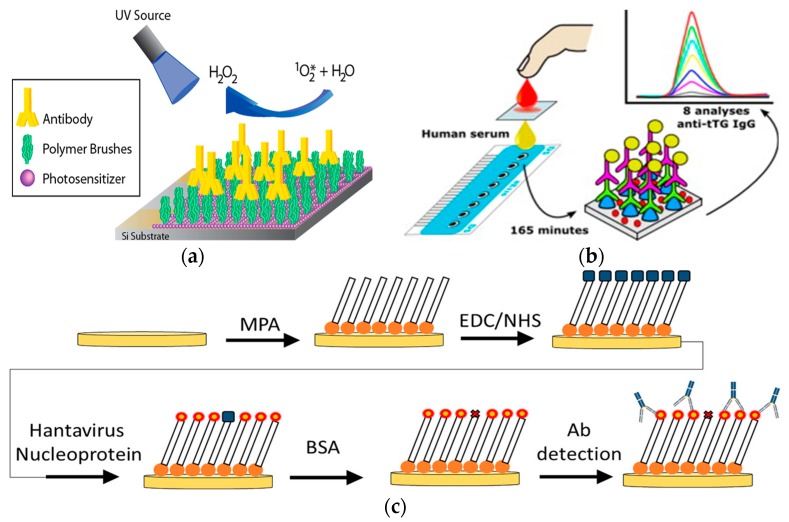
(**a**) Schematic of antibody-catalyzed water oxidation pathway-based electrode. (Adapted with the permission from Ref. [33]. Copyright © 2014 American Chemical Society.) (**b**) Schematic of the biosensor for detection of the antibody-bound quantum dots. (Adapted from Ref. [38]. Copyright © 2014 Elsevier B.V.) (**c**) Illustration of the fabrication process of self-assembled monolayer with immobilized hantavirus nucleoprotein for detection of the antibody (Adapted from Ref. [44]. Copyright © 2019 Elsevier B.V.).

### 2.3. Anti-Fouling Detection Strategy

An electrode is commonly used as a sensing element for signal transduction. Therefore, the natural affinity of metal electrodes with biomolecules can lead to a fouling issue, affecting the performance of the biosensor in sensitivity, detection limit, and accuracy [24,25,27,45]. Thus, it is critical to minimize the fouling effects. Using the antifouling characteristic of cysteamine-modified graphene oxide interface, a recent study demonstrated a highly sensitive method for auto-antibody detection in undiluted human serum (Figure 4a) [46]. The recognition element was *α*-Syn antigen covalently attached to the graphene oxide carboxylic groups. The transduction method measured the impedance change induced by the target antibody with non-faradaic EIS analyses. The sensor surface was prepared by immersing a cysteamine-modified gold electrode in a graphene oxide solution. The electrode surface possessed non-fouling characteristics due to its highly hydrophilic nature and the hydroxyl, epoxy and carboxyl groups in graphene oxide enhancing the detection of the target antibodies. This detection processed a low detection limit of 1.2 pM and a high selectivity in undiluted human serum.

A graphenyl surface was also used to improve the antibody detection (Figure 4b) [47]. A highly sensitive biosensing platform was developed to detect glutamic acid decarboxylase antibodies (GADA). The recognition element was GAD antigens linking onto the carboxylated-graphene modified gold surface. The transduction was the current signal change measured with ferricyanide reagent. Compared with a self-assembled monolayer with MPA, the carboxylated-graphene modified surface possessed a lower detection limit and a wider dynamic range for antibody detection. To eliminate the matrix effect, A/G protein modified magnetic beads were employed to isolate GADA antibodies from human serum due to the binding sites on A/G protein. With the presence of GADA antibodies bound to the A/G magnetic beads, the GAD antigens on the biosensor attached to the graphenyl surface captured the antibodies leading to insulation on the electrode resulting in lower redox current of the ferri/ferrocyanide probe. The detection limit was in the pico-molar range. 

A magnetic core-shell Fe_3_O_4_ nanoparticles-based electrochemical biosensor was developed addressing the fouling issue (Figure 4c) [48]. The recognition element was brucella outer membrane protein 31 (OMP31). Electrochemical signals were measured using DPV with Ru(NH_3_)_6_^3+^/KCl solution as the transduction mechanism. The sensor surface was prepared by adding Fe_3_O_4_–gold nanoparticles with polyethylene glycol and hyaluronic acid to a glassy carbon electrode. The limit of detection of the biosensor was 0.36 fg/mL. Testing of the sensor in blank serum, helper phage, and cow milk indicated less than 3.05%, 1.0%, and 1.27%, respectively, of non-specific protein adsorption by the modified Fe_3_O_4_–Au–PEG–HA nanoparticles. With the high sensitivity detection at concentrations significantly below typical blood serum antibody levels, this strategy showed potential as an early diagnostic method. 

The strategies shown resolve the fouling effect, preventing non-specific adsorption on the electrode surface. The first strategy reduced fouling effect on the electrode by modifying the graphenyl surface. The second strategy used the matrix effect through protein A/G linked MAG beads. Additionally, modifications to the biosensors using magnetic core shell Fe_3_O_4_ effectively prevented non-specific interactions with other elements of the medium while retaining selectivity with the target antibodies. The advances made in preventing non-specific adsorption to the electrode allow accessible testing in the presence of other biomolecules.

### 2.4. Rapid Detection Strategy

Clinical diagnosis with a rapid turnaround time enhances the time-efficiency of the healthcare workflow [14,49]. An effective method to detect HIV antibodies in complex biological samples based on steric hindrance was recently developed (Figure 5a) [50]. The recognition element was the epitope of gp41 attached to a peptide nucleic acid (PNA) sequence. The transduction method involved the current signal of the methylene blue tag induced by the efficiency of hybridization through the steric hindrance effect depending on the presence or absence of the antibodies. A 3D nanostructured gold was deposited on an indium tin oxide (ITO) electrode. The peptide, epitope of gp41, was attached to the PNA sequence hybridized to a 32-base-long DNA signaling strand forming a sensing complex. The epitope was bound to the target antibodies leading to changes in hybridization between the signal DNA strand and the capture DNA strand on the electrode through the steric hindrance effect. The redox label, methylene blue, was conjugated to one end of the signaling DNA strand. With the presence of antibodies on the peptide-PNA-DNA sensing complex, the steric hindrance effect reduced the number of redox labels bound to the electrode resulting in a reduced current signal. The detection limit was nanomolar and the readout signal was ten times larger than those of the other electrochemical sensors in less than ten minutes.

Rapid detection is also important for allergic reaction screening. Passive antibody therapy is an effective treatment as an antivenom. Prado et al. investigated an electrochemical sensor that detected the reactive IgE antibodies responsible for the allergic reaction rapidly (Figure 5b) [51]. The recognition element was goat anti-human IgE antibody with alkaline phosphatase. Cyclic voltammetry (CV) quantified the current changes through the redox reaction of hydroquinone by the alkaline phosphatase as the transduction mechanism. The sensor surface was prepared by crosslinking horse IgG3 (hoIgG3) to a chitosan film on the gold or carbon electrode using glutaraldehyde. The limit of detection for this sensor was 0.5 pg/mL. This strategy provided a rapid screening to determine potential patient allergies to passive antibody therapy before administering any other treatment.

Organic thin-film transistor (OTFT) was also developed for rapid antibody detection. (Figure 6a) [52]. This device also quantified the binding affinity between the antibody and the antigen. A modified pentacene-based organic thin-film transistor (OTFT) was used as a real-time and selective immuno-detection platform for antibody detection. The sensing element was a bottom-contact OTFT with a dielectric layer, source-drain gold electrodes, and a pentacene layer. A perfluor-1,3-dimethylcyclohexan (PFDMCH) layer was used to passivate the OTFT surface and a maleic anhydride (MA) layer was used to chemically functionalize the OTFT surface. The ppMA layer was activated with EDC/NHS (1-ethyl-3-(3-dimethylaminopropyl)carbodiimide/N-hydroxysuccinimide) on which a catcher probe was covalently attached as the recognition element. The transduction mechanism was a titration flow system crossing the surface of the sensor using a flow cell. As the target antibody concentration changed, the current output of the flow cell shifted accordingly. This platform detected antiBSA using BSA as the catcher probe and detected antibody at nanomolar levels. This OTFT platform was effective for antibody detection, and also for the quantification of bonding strength and charge discrimination capabilities. Comparing with electrochemical devices for quantification of biomolecular binding affinity [53], the direct transduction of binding affinity to electrical signal through OTFT demonstrated a potential universal platform for biophysical property analysis. 

These strategies used various methods to reduce the turnaround time for antibody detection. Enhanced steric hindrance with a 3D nanostructured electrode was applied to rapid HIV antibody detection. The use of biosensor detection for antibody therapy screening is another potential application for rapid detection sensors, as shown by creating sensors for detection in human serum of the reactive antibody IgE. Additionally, thr OTFT platform illustrates the potential and the rapid nature of a real-time biosensing platform. 

### 2.5. Low-Cost Detection Strategy

Inexpensive antibody detection is a focus for electrochemical-based biosensors. A low-cost electrochemical assay for the detection of anti-hepatitis C virus (HCV) was developed using hydrolysis reaction [54]. This strategy used genetically engineered yeast bio-bricks that display HCV core protein concatenated to gold-binding peptide (GBP) for a biosensor (Figure 6b). The sensing element was a screen-printed gold electrode with immobilized HCV core protein/GBP located on the surface of the yeast cell and mouse anti-HCV-core IgG antibody bound to the HCV core protein. The recognition element was a rabbit anti-mouse IgG antibody with alkaline phosphatase. The transduction mechanism was the electrochemical signals generated in the conversion of p-aminophenyl phosphate (pAPP) to p-aminophenol (pAP), measured through CV utilizing a potentiostat connected through the audio port of a smartphone. This strategy used surface-expressed proteins on renewable bio-bricks in electrochemical sensing in conjunction with the readily accessible and available smartphone to capture readings, potentially creating a more affordable point of care diagnostics. The reduced cost of antibody detection in the strategy is achieved through the use of reusable detection platforms and accessible detection devices. 

## 3. Optical Detection

Optical-based biosensors are also widely developed for antibody detection [13,55]. High-sensitivity quantification through optical imaging are achieved within seconds [56,57] Moreover, optical imaging is more suitable for the application of in-vivo sensing [58,59], owing to its direct signal translation process. Herein, we review recent developments of optical sensing strategies, demonstrating a great potential for antibodies screening or detection system. 

### 3.1. Rapid Detection Strategy

Using the base-pairing chemistry of DNA, optical-based biosensors allow for accurate and rapid antibody detection. Recent research demonstrated a sensitive, and highly specific method to detect antibodies (Figure 7a) [60]. The recognition element was the epitope or corresponding antigen to the antibody of interest. The transduction method was the target antibody-induced generation of a fluorescent signal. A programmable DNA nano-switch was designed with three individual strands of nucleic acids: the first strand was a stem loop structure consisting of a pair of fluorophores and quencher with a 15-base-long toehold region; the second strand was a recognition element conjugated ssDNA complementary to the toehold region of the first strand, forming a reporter module duplex; and the third strand with the same recognition element was hybridized to the loop region of the first strand. The sequence design of the first strand and second strand allowed hybridization only in close proximity sterically. With the presence of the antibody, the reporter duplex and strand 3 were bound to the antibody scaffold, initiating the strand displacement reaction with the release of the fluorescence signal. The detection limit of this strategy was at the nanomolar level. The turnaround time of this platform was around 15 min.

Similarly, a rapid and programmable platform for antibody detection using a DNA-based beacon was developed (Figure 7b) [61]. The recognition elements were small molecules, polypeptides, and epitopes corresponding to different target antibodies. The transduction element was the emission of the fluorophore/quencher pair on the DNA stem-loop structure induced by the target antibody. The two single strands of DNA each contained one recognition element based on the design of the structure. This platform detected one or two bivalent molecules, and even as the logic gates for the detection of two multivalent molecules. For example, it detected antiDig antibodies with digoxigenin (Dig) on each side of the strand. It also detected monovalent TATA binding protein (TBP) with hairpin DNA corresponding to the TBP sequence. Moreover, by changing the design of the recognition element to Dig and DNP on two strands, this sensor detected both the presence of anti-Dig and anti-DNP antibodies. With the presence of the targets, the fluorescence signal was generated by the opening of the stem-loop structures induced by the hybridization of the target to the recognition element. The detection limit reached nanomolar concentrations and the detection time was lowered to 2 min. 

These methods are built on the construction of DNA hairpin structures and conformational changes induced signaling mechanism, and they allowed for rapid detection. Due to the selective and rapid antibody-antigen binding process, target antibodies were optically detected using fluorescence/quencher pairs in less than 10 min. Shortening of detection times using structure-switching design in antibody detection satisfies the need for rapid detection of disease in point-of-care systems. 

### 3.2. Sensitive Detection Strategy

High sensitivity of biosensor is needed for antibody detection. Gold-coated nanoparticles were used to enhance its sensitivity in antibody detection (Figure 8a) [62]. A modified optical fiber long period grating (LPG) linked onto gold-coated biosensors was developed recently. The sensing element was an LPG biosensor using three layers of poly(allylamine hydrochloride)/gold-coated silica nanoparticles. The recognition element was a chemically functionalized layer on the nanoparticles that directly bound to the antibodies. Furthermore, the functionalized nanoparticle layer was attached a hapten. In one case, the nanoparticles were decorated with amine groups to attach the hapten biotin for streptavidin detection. The platform also utilized EDC/NHS enabling the covalent binding of anti-IgM to activate esters. To quantify the antibody population, the transduction element was characterized by changes in refractive index of the LPG, creating shifts in the wavelength of light traveling through the sensor. The changes quantified the antibody detected based on their direct proportional relationship to the antibody adsorption. 

A peroxidase-mimetic nanomaterials-based sensor was demonstrated recently (Figure 8b) [63]. The recognition element was the antigen attached to the carbon electrode. The transduction method was colorimetric and an electrochemical signal change induced by the oxidation reaction of TMB$H_{2}O_{2}$ catalyzed by the IgG/-NP${\mathrm{Fe}}_{2}O_{3}$NC. Gold-loaded superparamagnetic ferric oxide nanotubes (NP${\mathrm{Fe}}_{2}O_{3}$NC) were used for a molecular sensor based on its enhanced peroxidase-like activity and highly paramagnetic feature. After the target antibodies bound to the antigens on the carbon electrode, the ɑ-IgG attached NP${\mathrm{Fe}}_{2}O_{3}$NC was captured by the antibodies. The catalyzed oxidation reaction of TMB solution in the presence of $H_{2}O_{2}$ produced a blue-colored transfer complex which turned into a yellow-colored product with the addition of a reaction stopping solution. Both absorbance and the current signal quantified the concentration of target antibodies. The use of NP${\mathrm{Fe}}_{2}O_{3}$NC provided a sensitive detection method with a detection limit of 0.08 U/mL.

These strategies illustrate the use of novel nano-scale materials to enhance the sensitivity of biosensors. The LPG platform is highly modifiable with the gold nanoparticles, while simultaneously offering sensitive detection through its wavelength-based transduction mechanism. Furthermore, the large surface of gold-loaded nano-porous ferric oxide nano-cube and the presence of gold nanoparticles in the NPNC platform enhance the TMB/H_2_O_2_ reaction, resulting in higher sensitivity of this detection strategy. Increased biosensor sensitivity in antibody detection can alleviate current challenges presented due to the small amount of target antibodies in blood/serum samples in the early stages of diseases.

### 3.3. Low-Cost Detection Strategy

The cost of antibody detection by optical biosensors is lowered through employing inexpensive material in the detection system. Microfluidic paper-based analytical devices (μPADs) employing bioluminescence resonance energy transfer (BRET) switches were developed for antibody detection (Figure 9a) [64]. The recognition element consisted of LUMinescent AntiBody Sensing proteins (LUMABS) on BRET sensors/switches. The transduction mechanism was a bioluminescent signal with a hue-based readout. The sensing element was a multilayered 3D-μPAD. This approach enabled a simplified fabrication process because the combination of inexpensive material and lowered precision still offered proper functionality. In the absence of target antibodies, the BRET switch was in a green light-emitting “closed” state. In the presence of target antibodies, the binding triggered conformational changes in the switch enabling a blue light-emitting “open” state. The signal was then collected and quantified using a digital camera. This platform detected antiHIV1, anti-HA, and anti-DEN1 at nanomolar levels and was effective as a simplified antibody detection platform. The combined use of μPADs and BRET switches simplified the fabrication process and simultaneously enabled the detection of three different antibodies. 

Quantum dot (QD) is a novel material for optical detection [21]. However, using quantum dots as electrochemiluminescence (ECL) emitters for biosensing applications is restricted by the use of toxic or rare materials. This strategy uses tin disulfide, a fullerene-like n-type semiconductor, as a nanomaterial for ECL biosensors (Figure 9b) [65]. The recognition element was an amino-modified assistant probe (AP) crosslinked with chitosan. The transduction mechanism occurred through the reaction between the S_2_O_8_^2−^ solution and the tin disulfide quantum dots (SnS_2_ QDs). The sensing element was a ternary system of SnS_2_ QDs with silver nanoflowers (AgNFs) immobilized onto glassy carbon electrodes. The sensor was in an “on” state when the sensor containing the crosslinked AP produced a strong ECL signal, and moved to an “off” state when DNA strands modified with Fc were bound to the sensor. Anti-CMV pp65 introduced to a multi-functionalized oligonucleotide-CMV pp65 peptide conjugate capture probe initiated a series of processes that created a large quantity of mimic target (MT) sequences. When the MT was captured by the AP, the modified DNA-Fc was released from the surface of the electrode, reverting the sensor back to the “on” state. This ECL biosensor achieved an antibody detection limit of 0.33 fM and used a more accessible and safer material for quantum dots biosensors. 

These strategies describe the widened applications along with low-cost sensing platforms. The μPADs platform provided a low-cost alternative for disease detection through the use of inexpensive materials in the form of a paper-based system and inexpensive quantification methods in the form of a digital camera. Furthermore, a quantum dot sensor provided an alternative to standard costly quantum dots by utilizing abundant and lower cost tin disulfide as potential material for the sensors. This approach can lead to the development of more affordable and accessible biosensors for point-of-care systems, which may be important in low-resource areas. 

## 4. Conclusions

Electrochemical and optical biosensing strategies show great potential for the development of antibody-detection platform technologies. Both electrochemical and optical detection platforms are attractive for future antibody detection due to factors such as rapid readout, cost-effectiveness, sensitivity, selectivity, and portability. Despite biosensing strategies having been demonstrated for antibody detection, there is still an unmet need for the development of antibody detection. An integration of current biosensor platforms into routine clinical usage still requires significant advancement in future research. To constuct a universal detection strategy, we need to integrate electrochemical and optical transduction mechanisms with the merits of different biosensing strategies. We believe that establishing the clinical protocols for sample processing is critical to generalized sample analysis. Furthermore, the integration of detection systems is limited and will require further development. Therefore, research on the integration of sensing elements, recognition elements and transduction elements into a preservable biosensor product requires more scientific and technological efforts to lead to a portable point-of-care system which is widely applicable globally. Moreover, advances in multiplex, in-vivo biosensing strategies remain limited. Once the technical challenges, such as advancements in biocompatible materials and in-vivo transduction of multiple signals are resolved, next-generation biosensing platforms can become a widespread standard for clinical point-of-care screening and detection systems. 

## Figures and Tables

**Figure 1 ijms-21-00134-f001:**
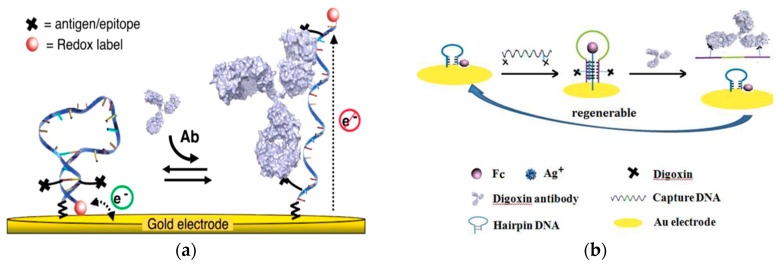
(**a**) Illustration of the electrochemical “switch” composed of single-strand DNA with linked antigens and methylene blue redox label. Introduction of the target antibody opens the loop and moves the redox label away from the electrode. (Adapted with permission from Ref. [28]. Copyright © 2012 American Chemical Society). (**b**) Schematic of the “triplex-stem” DNA molecular switch on gold electrode. Addition of capture DNA (middle) shows a ferrocene molecule moving away from the electrode, and digoxin antibody introduced to the system binds with digoxin on the ends of the capture DNA and causes it to be released, converting the triplex system to hairpin DNA with ferrocene close to the electrode surface. (Adapted from Ref. [29]. Copyright © 2014 Elsevier B.V.)

**Figure 2 ijms-21-00134-f002:**
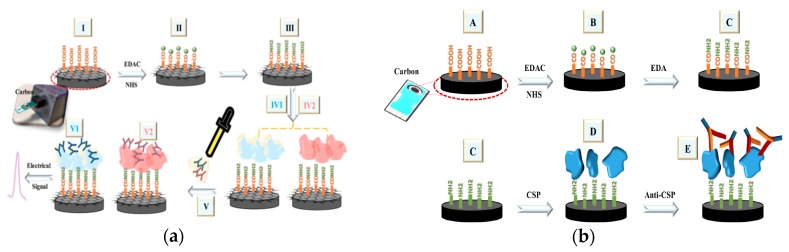
(**a**) Schematic of the fabrication of the Zika virus (ZIKV) antibody-detecting biosensor by activation of carboxylic acid ends (II), and incubation with NS1 and EDIII protein (IV1 and IV2, respectively). (Adapted from Ref. [30]. Copyright © 2018 Elsevier B.V.) (**b**) Schematic of the fabrication of the circumsporozoite (CSP) antibody detecting biosensor. Carboxylic acid ends are activated (B), aminated by the addition of ethylenediamine (C), incubated with CSP protein 210-247 (D), and bound to Anti-CSP 210 or 247 (E). (Adapted from Ref. [31]. Copyright © 2017 Elsevier B.V.)

**Figure 4 ijms-21-00134-f004:**
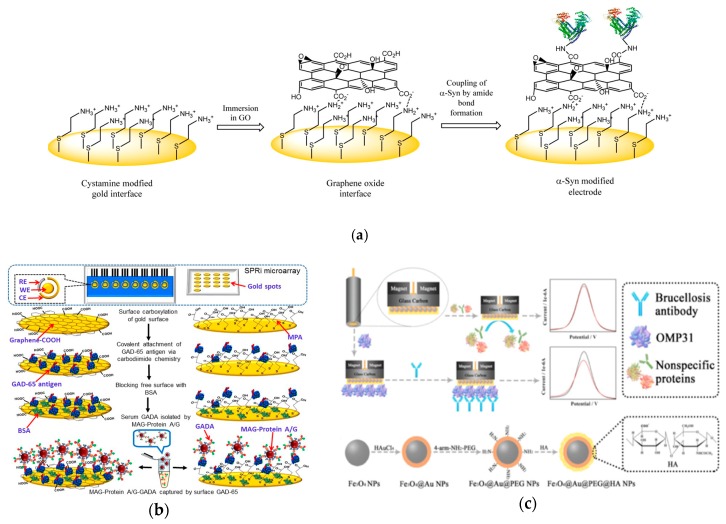
(**a**) Schematic illustration of the sensing platform. Cysteamine-modified gold interface is immersed in graphene oxide and then activated to form amide bonds with ɑ-Syn antigens (Adapted with permission from Ref. [46] Copyright 2015 American Chemical Society). (**b**) Comparative analysis between graphenyl surface and mercapto-monolayer surface. (Adapted with permission from Ref. [47]. Copyright 2018 American Chemical Society.) (**c**) Top: schematic of electrochemical sensor for detection of brucellosis antibody on magnetic glassy carbon electrode. Bottom: schematic for formation of magnetic core-shell Fe3O4 nanoparticles modified with polyethylene glycol and hyaluronic acid. (Adapted from Ref. [48]. Copyright © 2018 Elsevier B.V.)

**Figure 5 ijms-21-00134-f005:**
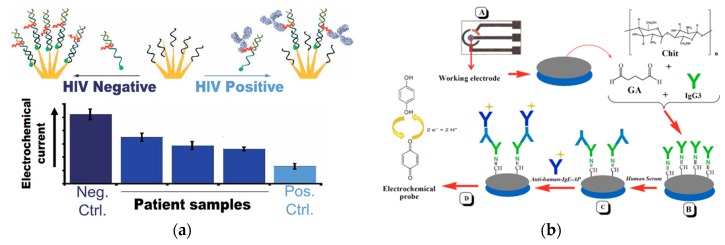
(**a**) With the presence of target antibodies, the hybridization of the redox-labeled PNA-peptide complex to the capture DNA bound to the electrode is reduced; the current signal is measured with the presence (black line) or absence (blue line) of the target antibodies (adapted with permission from Ref. [50] Copyright 2019 American Chemical Society). (**b**) Schematic of the electrochemical sensor for detection of IgE. Carbon or gold electrode (A) is covered with a chitosan film through combination of glutaraldehyde and hoIgG3 (B). Test serum is added to capture human IgE (C), and goat anti-IgE labeled with alkaline phosphatase is added to enable the reaction with hydroquinone (D). (Adapted from Ref. [51]. Copyright © 2017 Elsevier B.V.)

**Figure 6 ijms-21-00134-f006:**
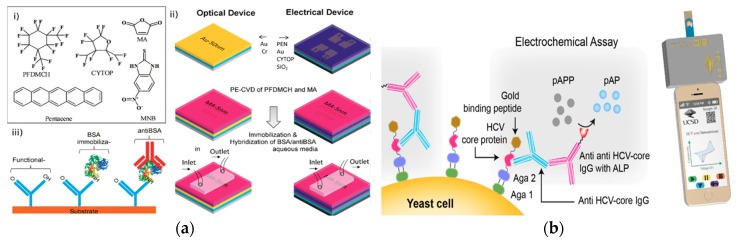
(**a**) (i) Organic molecule chemical structures. (ii) Schematic of the fabrication process of an Organic Thin Film Transistor (OTFT) with source-drain gold electrode, dielectric layer, and pentacene layer for passivation and functionalization of the OTFT surface. (iii) Schematic of the sensor to selectively detect anti-BSA. (Adapted with permission from Ref. [52]. Copyright © 2014 American Chemical Society.) (**b**) Schematic of the expressed HCV core protein bound to HCV-core IgG antibody, with electrochemical detection of the anti-HCV-core IgG antibody through the reaction of p-aminophenyl phosphate (pAPP) with alkaline phosphatase (ALP). (Adapted from Ref. [54]. Copyright © 2016 Elsevier B.V.)

**Figure 7 ijms-21-00134-f007:**
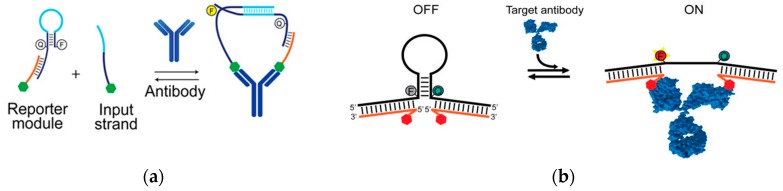
(**a**) With the presence of the target antibody, the reporter module and input strand are in close proximity with each other which brings up the efficiency of hybridization. The binding of the recognition elements and the target antibodies leads to the release of the signal from the fluorescence/quencher pair. The signal gain is recorded with different concentrations. (Adapted with permission from Ref. [60] Copyright 2018 American Chemical Society.) (**b**) The binding of the target antibody to the recognition elements on two single-stranded tails opens the stem-loop structure, leading to the release of the fluorescence signal (adapted from Ref. [61], copyright 2015 WILEY-VCH Verlag GmbH & Co. KGaA, Weinheim).

**Figure 8 ijms-21-00134-f008:**
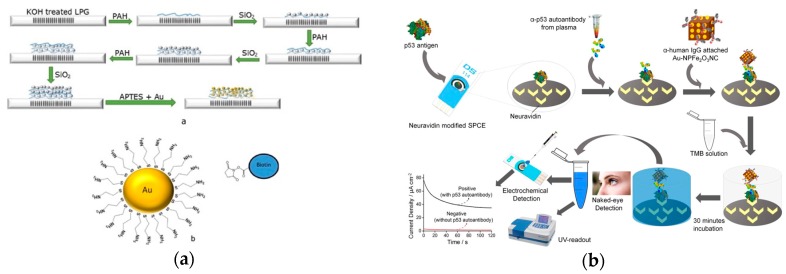
(**a**) Top: Illustration of the fabrication process for a long period grating (LPG) biosensor through the addition of three layers of poly-allylamine hydrochloride (PAH) and silica nanoparticles, functionalized with gold nanoparticles. Bottom: gold nanoparticle structure that can bind biotin. (Adapted from Ref. [62]. Copyright © 2018 Elsevier B.V.) (**b**) Illustration of antibody detection using Au-nanoparticles using optical and electrochemical methods. (Adapted with permission from Ref. [63]. Copyright 2017 American Chemical Society.)

**Figure 9 ijms-21-00134-f009:**
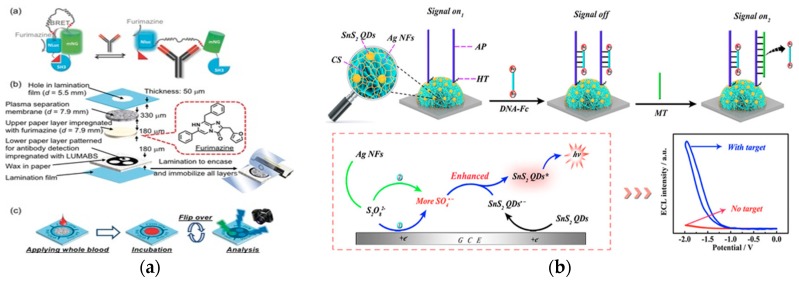
(**a**) Top: schematic of green light-emitting luminescent antibody-sensing proteins (LUMABS) in the absence of target antibody and blue light-emitting LUMABS in the presence of antibody. Middle: schematic of a microfluidic paper-based analytical device. Bottom: schematic of detection of three separate antibodies using a single device. (Adapted from Ref. [64]. Copyright © 2018, The Authors, published by Wiley-VCH Verlag GmbH & Co. KGaA.) (**b**) Top: schematic of the SnS2 quantum dots and Ag nanoflowers ternary structure on the glassy electrode and subsequent signal off and on mechanism. Bottom: Proposed ECL mechanism of the system. (Adapted with permission from Ref. [65]. Copyright 2018 American Chemical Society.)

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
