# Peer review of "Recent Developments of Electrochemical and Optical Biosensors for Antibody Detection"

_ijms, 2019, doi:10.3390/ijms21010134_

Round 1

Reviewer 1 Report

In this manuscript, the authors review the recent applications of electrochemical and optical biosensors for antibody detection. The recent developments in the electrochemical and optical biosensors are described, with a focus on selectivity, sensitivity, and cost-effectiveness of the proposed devices and sensing platforms. The topic is of interest in the field of point-of-care devices.

The manuscript can be considered for publication after minor revision.

A table summarizing the main developments and improvements in the analytical performances of the electrochemical and optical biosensors should be provided. The transduction mechanism for electrochemical biosensors using a redox probe like ferro/ferricyanide is actually based on measurements of the faradic current and/or of the impedance of the electrode/solution interface. Therefore, sentences like “The transduction element was [Fe(CN)6]3-/4- based redox system” and „A redox solution of K3Fe(CN)6 and K4Fe(CN)6 was the ...” should be avoided. The title “Rapid Detection Strategy” is repeating for sections 2.4 and 2.5, respectively.

Minor points:

The acronyms should be defined after their first usage.

Author Response

Thank you very much for your kind advice. 

We have defined the acronyms for first time usage as advised throughout the text.

The subtitle 2.5 has been modified to low cost detection strategy as suggested.

We also deleted the sentence " The transduction element was the [Fe(CN)63-/4] based redox system and "a redox solution of K3FeCN6..." as advised.

Many thanks  

Reviewer 2 Report

The manuscript presents a review on antibody detection strategies. The manuscript is of interest for researchers who need to develop biosensors based on antibody quantification, particularly in sample sera. However, the manuscript should be modified before being considered for publication. Hoping that this contribution can help to improve the manuscript, I release herein my comments:

Major revisions

- The abstract does not describe properly the content inside the manuscript. I strongly suggest modifying it including the critical points discussed inside the as for example the detection strategies included . In the most part of the cases, authors described and discussed antibody-detection strategies, not diagnostic assays o trials.

-I suggest to remove the phrase (line 34-35) in which authors wrote “…, antibodies, which are blood circulating proteins produced by the immune system, …” The readers target of this journal should be able to know it.

-Authors could highlight the problematic linked to the fact that antibodies should be detected directly on sera, particularly in the case of electrochemical detection working with gold electrodes. In fact, the most part of the literature included in the manuscript described methods which were set-up using purified antibodies, and lack on papers working directly with sample sera (i.e. Clin Chim Acta. 2015;444:199-205, Electrochimica Acta. 2015, 176, 1239-1247).

-Authors presented an incomplete selection on recent literature; they should included the terms in which papers were selected for the review, and the areas they pretended to cover with.

-I suggest modifying the title of running head 2.5 “rapid detection strategy” for other one taking into account that the discussion is based on costs and materials. Nothing is discussed on terms of detection times, as in the previous paragraph.

-In the conclusions, authors claim that the detection systems are not integrated and require further development (lines 492-493). This point is not discussed inside, the review manuscript could be highly improved if a brief comment regarding unmet challenges regarding applications is included.

Minor revision

-In the abstract line 13 authors reported “…in the future point-of-care system.” Please use the plural form “systems”.

Author Response

Thank you very much for your kind advice

We have modified the manuscript as advised Specifically

Suggestion 1

the abstract does not describe properly the content inside the manuscript. I strongly suggest modifying it including the critical points discussed inside the as for example the detection strategies included . In the most part of the cases, authors described and discussed antibody-detection strategies, not diagnostic assays o trials.

We have added the specific details of detection strategies in the abstract as 

In this review, electrochemical and optical detections of antibody are discussed. Specifically, creating a non-label and reagent-free sensing platform and construction of an anti-fouling electrochemical surface for electrochemical detection are suggested. For optical transduction, a rapid and programable platform for antibody detection using DNA-based beacon is suggested as well a the use of bioluminescence resonance energy transfer (BRET) switch for low cost antibody detection is included in this review.

Suggestion 2

-I suggest to remove the phrase (line 34-35) in which authors wrote “…, antibodies, which are blood circulating proteins produced by the immune system, …” The readers target of this journal should be able to know it.

We have deleted the sentence as suggested.

Suggestion 3

-Authors could highlight the problematic linked to the fact that antibodies should be detected directly on sera, particularly in the case of electrochemical detection working with gold electrodes. In fact, the most part of the literature included in the manuscript described methods which were set-up using purified antibodies, and lack on papers working directly with sample sera (i.e. Clin Chim Acta. 2015;444:199-205, Electrochimica Acta. 2015, 176, 1239-1247).

We have added in details regarding detection of antibodies directly in whole blood and serums in the original text . We also added references from the above two publications in the reference lists [17,18]

Suggestion 4

-Authors presented an incomplete selection on recent literature; they should included the terms in which papers were selected for the review, and the areas they pretended to cover with.

We recognized that we cannot cover all the literature and selected the few publications that we think can be representative. We have added a sentence in the text expressing this issue.

Suggestion 5

-I suggest modifying the title of running head 2.5 “rapid detection strategy” for other one taking into account that the discussion is based on costs and materials. Nothing is discussed on terms of detection times, as in the previous paragraph.

We agreed with the suggestion and has modified the subtitle 2.5 to "low cost detection strategy"

Suggestion 6

In the conclusions, authors claim that the detection systems are not integrated and require further development (lines 492-493). This point is not discussed inside, the review manuscript could be highly improved if a brief comment regarding unmet challenges regarding applications is included.

We have modified the conclusion stating the unmet challenge of antibody detection would require an integration of both electrochemical and optical detection methods with biosensing strategies. 

Suggestion 7(minor)

-In the abstract line 13 authors reported “…in the future point-of-care system.” Please use the plural form “systems”.

We have made the modifications based on the suggestion.

Round 2

Reviewer 2 Report

The manuscript has been modifyed following the reviewer comments. I suggest to publish in the present form.